# Anaerobic Gram-Negative Bacteria: Role as a Reservoir of Antibiotic Resistance

**DOI:** 10.3390/antibiotics12050942

**Published:** 2023-05-22

**Authors:** Anshul Sood, Pallab Ray, Archana Angrup

**Affiliations:** Department of Medical Microbiology, Post Graduate Institute of Medical Education and Research (PGIMER), Chandigarh 160012, India; anshulsood02@gmail.com (A.S.); drpallabray@gmail.com (P.R.)

**Keywords:** antimicrobial resistance, Gram-negative anaerobic bacteria, *Bacteroides* spp., *Fusobacterium* spp., *Prevotella* spp., *Veillonella* spp.

## Abstract

Background: Anaerobic Gram-negative bacteria (AGNB) play a significant role as both pathogens and essential members of the human microbiota. Despite their clinical importance, there remains limited understanding regarding their antimicrobial resistance (AMR) patterns. This knowledge gap poses challenges in effectively managing AGNB-associated infections, as empirical treatment approaches may not adequately address the evolving resistance landscape. To bridge this research gap, we conducted a comprehensive study aimed at exploring the role of human AGNB as a reservoir of AMR. This can provide valuable insights for the prevention and management of anaerobic infections. Methods: We studied the prevalence of AMR and AMR determinants conferring resistance to metronidazole (*nimE*), imipenem (*cfiA*), piperacillin–tazobactam (*cepA*), cefoxitin (*cfxA*), clindamycin (*ermF*), chloramphenicol (*cat*) and mobile genetic elements (MGEs) such as *cfiA^IS^* and *IS*1186 associated with the *cfiA* and *nim* gene expression. These parameters were studied in *Bacteroides* spp., *Fusobacterium* spp., *Prevotella* spp., *Veillonella* spp., *Sutterella* spp., and other clinical AGNB. Results: Resistance to metronidazole, clindamycin, imipenem, piperacillin–tazobactam, cefoxitin and chloramphenicol was 29%, 33.5%, 0.5%, 27.5%, 26.5% and 0%, respectively. The presence of resistance genes, viz., *nim*, *ermF*, *cfiA*, *cepA*, *cfxA,* was detected in 24%, 33.5%, 10%, 9.5%, 21.5% isolates, respectively. None of the tested isolates showed the presence of a *cat* gene and MGEs, viz., *cfiA^IS^* and IS*1186*. The highest resistance to all antimicrobial agents was exhibited by *Bacteroides* spp. The association between resistant phenotypes and genotypes was complete in clindamycin, as all clindamycin-resistant isolates showed the presence of *ermF* gene, and none of the susceptible strains harbored this gene; similarly, all isolates were chloramphenicol-susceptible and also lacked the *cat* gene, whereas the association was low among imipenem and piperacillin–tazobactam. Metronidazole and imipenem resistance was seen to be dependent on insertion sequences for the expression of AMR genes. A constrained co-existence of *cepA* and *cfiA* gene in *B. fragilis* species was seen. Based on the absence and presence of the *cfiA* gene, we divided *B. fragilis* into two categories, Division I (72.6%) and Division II (27.3%), respectively. Conclusion: AGNB acts as a reservoir of specific AMR genes, which may pose a threat to other anaerobes due to functional compatibility and acquisition of these genes. Thus, AST-complying standard guidelines must be performed periodically to monitor the local and institutional susceptibility trends, and rational therapeutic strategies must be adopted to direct empirical management.

## 1. Introduction

The emerging antimicrobial resistance (AMR) is a critical problem faced by the medical and scientific community. While extensive research has addressed AMR in aerobic and facultative anaerobic pathogens, the study on strict anaerobes has received comparatively less attention. Anaerobic Gram-negative bacteria (AGNB) comprise a significant proportion of the human microbiota and often act as secondary pathogens [1]. AGNB are the most common anaerobes associated with infections and include some of the most antimicrobial-resistant species [2]. These bacteria, particularly the *Bacteroides fragilis* group, have exhibited notable resistance rates [3], with some strains demonstrating multiple-drug resistance (MDR) [4]. MDR is now being recognized among other clinical AGNB as well, which were earlier believed to be susceptible [5]. Most of the available literature on AMR belongs to the genus *Bacteroides*, and the remaining AGNB continue to take a back seat. These bacteria can serve as channels for horizontal gene transfer (HGT) and the ignored conduct may lead to the possibility of selection and transfer of resistance determinants [6].

Understanding the extent of AMR in AGNB is essential for several reasons. AGNB are clinically significant, often associated with polymicrobial infections, and can cause severe infections such as abscesses, bacteremia, and intra-abdominal infections [2]. The appropriate management of these infections requires accurate antimicrobial susceptibility testing (AST) and an understanding of local resistance patterns. However, the current knowledge of AMR patterns in AGNB is limited, and the existing data often deviate from the standard guidelines [7,8]. The changing antibiograms and the emergence of resistance determinants in AGNB mandate periodic phenotypic and genotypic AST of these bacteria. Moreover, AGNB exhibit intrinsic resistance mechanisms that differ from those observed in aerobic bacteria. They possess distinct sets of resistance genes, including those encoding resistance to critical antibiotics [9], making effective treatment challenging. In AGNB, there has been a focus on studying AMR, especially genotypic resistance in limited species and against a restricted spectrum of antibiotics, many of which hold less therapeutic significance today.

In this study, we aimed to address these knowledge gaps and explore the prevalence of AMR among clinically significant AGNB. We assessed the phenotypic and genotypic resistance among clinically significant AGNB to at least one drug from each class of antimicrobial agents recommended by CLSI for the primary testing of anaerobes, viz., metronidazole, clindamycin, piperacillin–tazobactam, and imipenem; and to two supplemental drugs for the selective testing, viz., chloramphenicol and cefoxitin. Due to the comparable clinical efficacy and interpretive results, only one antimicrobial agent from each antimicrobial class was included [10]. We intentionally excluded all ß-lactams since most of the *B. fragilis* group members are reported as uniformly resistant to them [11]. In this study, the resistance to each antimicrobial agent was considered, including (a) phenotypic resistance, an isolate with a “resistant” phenotype, when the MIC was greater than the breakpoint, and the data are interpreted by including the intermediate and resistant phenotypes in the resistant category; and (b) genotypic resistance, an isolate with a “resistant” genotype, when the isolate harbored the gene encoding for AMR to the given antimicrobial agent. We also studied the prevalence of the AMR determinants for the aforementioned antimicrobials, viz., *nimE* (metronidazole), *cfiA* (imipenem), *cepA* (piperacillin–tazobactam), *cfxA* (cefoxitin), *ermF* (clindamycin), *cat* (chloramphenicol) and MGEs such as *cfiA^IS^*, the insertion sequence present upstream of *cfiA* gene known to upregulate the expression of the *cfiA* gene and resultant imipenem resistance, and *IS*1186, associated with the *cfiA* expression and in many cases known to induce *nim* gene-mediated metronidazole resistance.

Overall, the study findings may have important implications, with the potential to enhance clinical decision-making, inform antibiotic stewardship efforts, and shape infection control strategies tailored to AGNB-associated infections. It underscores the importance of addressing AMR challenges and improving patient outcomes.

## 2. Results

A total of 200 consecutive AGNB were included in the study; these isolates represented 11 genera and 38 species (Table 1). Anaerobic Gram-negative bacilli comprised the majority, 78.5% (157/200), and the remaining 21.5% (43/200) isolates belonged to Gram-negative cocci.

### 2.1. Phenotypic Resistance in Anaerobic Gram-Negative Bacteria

Minimum inhibitory concentration (MIC) distribution in AGNB to different tested antimicrobials is summarized in Appendix A, and Appendix A depicts the antimicrobial susceptibility of clinical isolates representing different genera to these antimicrobials. The clinical AGNB showed the highest resistance against clindamycin, 33.5% (67/200). Among the various species, *B. fragilis* showed the highest resistance at 46.8% (22/47) to clindamycin. A high resistance rate was noted for metronidazole, as 29% (58/200) of the tested isolates were metronidazole-resistant. The genus *Veillonella* exhibited the highest resistance for metronidazole at 48.6% (17/35), followed by *Bacteroides*, 41.1% (30/73). The miscellaneous group of Gram-negatives was completely susceptible to the drug except for *Sutterella* spp., where 7 of 12 *S. wadsworthensis* isolates were resistant. Metronidazole maintained good activity against genus *Fusobacterium* and *Prevotella*. The overall resistance to ß-lactams was as follows: piperacillin/tazobactam, 27.5% (55/200); imipenem, 0.5% (1/200) and cefoxitin, 26.5% (53/200). The highest resistance rate against piperacillin–tazobactam was seen in *B. fragilis,* 44.6% (21/47), and no resistance was observed in the genus *Prevotella*. The resistance against cefoxitin was relatively less intense and more distributed among different species. The highest resistance against cefoxitin was seen in the genus *Bacteroides,* 43.8% (32/73). The drug showed good activity against the miscellaneous AGNB and *Fusobacterium* spp. since only two isolates of *B. wadsworthia* and three isolates of *F. nucleatum* showed resistance to the drug. The result shows that AGNB remains the most susceptible to imipenem out of all ß-lactams. Only one isolate of *B. fragilis* showed intermediate resistance to the drug, making up the total resistance against imipenem at 0.5% (1/200). Chloramphenicol demonstrated excellent activity against the tested AGNB with 100% susceptibility across all species. However, the clustering of MICs was centered around the breakpoints (55% at 4 mg/L; 17.5% at 8 mg/L), as shown in Appendix A. No such clustering around the breakpoints was observed in other antimicrobials. The number of not susceptible isolates with the percentage resistance against selected antimicrobials and the occurrence of corresponding AMR genes in tested species of genus *Bacteroides*, *Fusobacterium, Prevotella, Veillonella,* and other AGNB are mentioned in Table 2, Table 3, Table 4, Table 5 and Table 6, respectively.

### 2.2. Genotypic Resistance in Anaerobic Gram-Negative Bacteria

The overall prevalence of AMR determinants was 24% (48/200) for *nim*; 33.5% (67/200) for *ermF*; 0% (0/200) for *cat*; 9.5% (19/200) for *cepA*; 21.5% (43/200) for *cfxA*; 10% (20/200) for *cfiA* gene and 0% (0/200) for IS*1186* and *cfiA^IS^*, insertion sequences present upstream of *cfiA* gene. The overall susceptibility results and distribution of resistance determinants in clinical AGNB are summarized in Table 7. The genotypic identification of AMR genes and their association with phenotypic resistance are summarized in Table 8, and the correlation is pictorially depicted in Figure 1. Figure 2 illustrates a separate correlation in *Bacteroides* spp. The agarose gel pictures of the amplified products (AMR determinants) are shown in Appendix A. Clindamycin displayed a complete association between phenotypic and genotypic resistance, where all intermediate and resistant strains showed the presence of *ermF* gene, 100% (67/67), and none of the susceptible strains harbored the gene (Figure 1). Similarly, for chloramphenicol, a 100% genotypic and phenotypic association was seen. The association was relatively lesser in cefoxitin, as one cefoxitin-susceptible strain was detected with *cfxA*, the gene encoding cefoxitin resistance, and 11/53 cefoxitin-resistant isolates were devoid of the *cfxA* gene. The non-associations were mainly seen in metronidazole, piperacillin–tazobactam, and imipenem. In metronidazole, 39.6% (23/58) resistant phenotypes lacked the *nim* gene, whereas 27% (13/48) isolates harbored the *nim* gene but showed susceptibility to the drug. The lack of association between resistant genotypes and phenotypes was seen mainly in *Veillonella* spp., where 48.5% (17/35) isolates showed resistance to the drug, but only 1/17 resistant isolates harbored the gene. Out of the eight *nim*-positive metronidazole-susceptible isolates, seven showed a reduced susceptibility and high MIC (8 mg/L). Of 27.5% (55/200) piperacillin–tazobactam-resistant isolates, 34.5% (19/55) carried the *cepA* gene; 12.7% (7/55) carried *cfiA* and 29% (16/55) carried *cfxA*. Out of 72.5% (145/200) piperacillin–tazobactam-susceptible phenotypes, none of the isolates carried the *cepA* gene, whereas 8.9% (13/145) isolates harbored the *cfiA* gene and 18.6% (27/145) harbored the *cfxA* gene. A total of 3.4% (5/145) piperacillin–tazobactam-susceptible isolates were detected with the presence of both the *cfiA* and *cfxA* genes; however, these isolates did not show reduced susceptibility to the drug. In 26.5% (53/200) cefoxitin-resistant isolates, the *cepA* gene was found in 7.54% (4/53) isolates; *cfiA* was found in 26.4% (14/53) and *cfxA* was found in 79.2% (42/53) isolates. Only one cefoxitin-susceptible isolate was detected with the *cfxA* gene, whereas *cepA* and *cfiA* were detected in 10.2% (15/147) and 4% (6/147) of cefoxitin-susceptible isolates, respectively. The 10.2% (15/147) *cepA*-positive cefoxitin-susceptible strains showed an increased MIC towards cefoxitin, viz., 11/15 at 16 µg/mL; 2/15 at 8 µg/mL. Similarly, all 4% (6/147) of *cfiA*-positive and 0.6% of (1/147) *cfxA*-positive cefoxitin-susceptible strains showed increased MIC at 16 µg/mL. The *cfiA* gene encoding imipenem-hydrolyzing metallo-ß-lactamase was found in 10% (20/200) of isolates, out of which one isolate showed intermediate resistance towards the drug; the remaining nineteen *cfiA*-positive isolates were imipenem-susceptible. No association was seen among imipenem-resistant phenotypes and genotypes, and the agreement was found insignificant with a *p*-value of >0.05. Further, we attempted to detect the insertion sequence elements immediately upstream of the *cfiA* gene, providing a promoter for the expression of the gene (Appendix AB). None of the isolates except the positive controls showed a positive band in the range of 1.6–1.7 kb, indicating the absence of IS elements, whereas bands were seen at 350 kb, demonstrating the presence of only *cfiA* genes in our isolates. A positive association and a significant *p*-value (<0.05) of IS elements were seen with the phenotypic resistance in imipenem (Table 8). Similarly, none of our clinical isolates were detected with *IS*1186, known to play a role in *nim*-induced metronidazole resistance. In this study, the ß-lactamases, *cfiA,* and *cepA* genes were limited to *B. fragilis* species, whereas the *cfxA* gene was relatively more widespread among other members of *Bacteroides* spp. and different AGNB (Figure 1). The results demonstrate a constrained co-existence of *cepA* and *cfiA* gene, as all the isolates positive with *cepA* gene lacked the *cfiA* gene and vice versa (Figure 3A). In addition, based on the absence and presence of the *cfiA* gene, we divided *B. fragilis* into two categories, Division I and Division II, respectively (Figure 3B).

## 3. Discussion

This is the first Indian study that describes the genotypic and phenotypic resistance in AGNB against six major anti-anaerobic drugs. In this study, we saw that AMR was most commonly found in *Bacteroides* spp. and was reported highest against clindamycin. Over the past 40 years, resistance against clindamycin has increased drastically worldwide, especially in the Asian countries. Data from the early 1980 and 1990s shows less than 5% resistance to clindamycin [12,13], which has increased to 32.4% after a decade, as reported in a Europe-wide study of 13 countries [14]. However, in the present scenario, resistance to clindamycin seems steady in the European countries [3,15,16,17], while it has increased to 50–68% in the US [18,19,20] and even higher in the Asian countries (80–90%) [15,21,22,23,24]. In our study, we detected a 33.5% resistance against clindamycin, and the results confirm a 100% association between resistant genotypes and phenotypes. The *ermF* gene was found as the clindamycin resistance determinant in our isolates, different from the prevalent variants known to confer clindamycin resistance in aerobes and Gram-positive anaerobes [8]. This may suggest that our clinical isolates have the potential to act as a reservoir of clindamycin resistance, certainly for AGNB. Our study reported a high resistance rate against clindamycin in the genus *Prevotella* (36.7%) and *Fusobacterium* (38.7%). The literature survey showed an even higher resistance rate in *Prevotella*, i.e., up to 40% in Europe, 50% in both US and Asia, and highest in Kuwait, around 89% [15]. In *Fusobacterium* spp., the resistance rates of Asian countries such as Taiwan (31%) [24] and Singapore (44%) [25] are similar to ours; however, a ≤15% resistance has been reported from the rest of the world. Clindamycin, which was earlier believed to be a drug of choice for infections above the diaphragm, now calls for attention, since *Prevotella* and *Fusobacterium* are usual isolates of orodental and other above-diaphragm infections.

Metronidazole can be used as a drug of choice to treat anaerobic infections caused by *Fusobacterium* and *Prevotella.* Unfortunately, resistance to metronidazole is also becoming a topic of concern [26]. Our study showed only one and three metronidazole-resistant isolates from these genera, respectively. In our isolates, a very low resistance was detected in *Fusobacterium* spp., which was similar to various studies worldwide with a few exceptions [27,28]. However, the literature shows an emerging AMR to metronidazole in the genus *Prevotella* [3] and *Veillonella* [29]. Reduced susceptibility of *Veillonella* isolates to metronidazole has been seen in our isolates (48.6%), which is in tune with the literature from the East Asian countries [28,30]. Metronidazole resistance among *B. fragilis* group isolates is emerging worldwide due to the non-judicious use of the drug to treat anaerobic infections. The European data from the early 1990s showed no resistance, but in the succeeding years, there was an increase to 0.5% [31,32,33]. On the contrary, a significantly higher resistance of up to 15% has been seen in many western countries [34,35,36] and up to 30% from a few Asian regions [22,35,36,37,38]. In our study, a 41.1% resistance was seen in *Bacteroides* spp., whereas resistance varying from 7% to 31% has been reported in other Indian studies [26,39]. The main mode of metronidazole resistance is nitroimidazole reductase activity encoded by *nim* genes [40]. The studies evaluating the presence of *nim* genes are limited; nevertheless, they depict a low prevalence ranging from 0.5 to 2.8% in *Bacteroides* spp. [27,29,40]; 0 to 5.3% in *Prevotella* spp. [27], and 0 to 5.9% in *Fusobacterium* spp. [27,40]. Yet again, most of the literature belongs to the European countries [40], and the geographic distribution of *nim* genes in the Indian subcontinent is relatively under-explored. In our study, the *nim* genes were detected in 24% (48/200) of isolates and were more prevalent in *Bacteroides* spp. at 49.3% (36/73). The findings were in accordance with an Indian study, where *nim* gene positivity was seen in 53% (20/38) of *Bacteroides* spp. [26]. Out of the 24% (48/200) *nim* gene-positive isolates, 27% (13/48) of isolates carried the *nim* genes, yet they were phenotypically susceptible, which is possibly due to the absence of IS elements that regulate the expression of the *nim* genes. Of all the IS elements reported so far, the isolates were tested for the presence of IS*1186*, which has been frequently studied in anaerobes [40]. Unexpectedly*,* all 200 isolates tested negative for IS*1186*, which shows that possibly other IS elements are prevalent in our geographic region, or alternative non-*nim*-induced mechanisms of metronidazole are more active in our isolates. The non-*nim*-gene-based mechanisms of metronidazole resistance such as overexpression of efflux pump, *RecA* proteins, rhamnose catabolism regulatory protein, activation of antioxidant defense systems, and deficiency of ferrous iron transporter feoAB have been described; however, the literature is scant [40]. In our study, the ideal possibility of elucidating the non-*nim*-gene-mediated resistance can be attributed to the *Veillonella* genus. The genus displayed the highest resistance to metronidazole yet showed the minimum number of resistant genotypes, as only 1 of 17 *Veillonella*-resistant isolates harbored the *nim* gene.

The emergence of metronidazole resistance has led to the use of carbapenems, particularly imipenem, for the treatment of anaerobes. Resistance to imipenem is due to a metallo-ß-lactamase encoded by the *cfiA* gene, which is expressed by the upstream IS element that carries a promoter to drive the gene expression [31]. Data obtained in the West from various studies reports an overall carbapenem resistance varying from 1 to 9.6% and *cfiA* positivity of 5 to 27% in *B. fragilis* [41]. The East Asian literature shows a 7% imipenem resistance in *B. fragilis,* 4% in *Fusobacterium* spp. over 16 years [24]. *B. fragilis* from China demonstrates a remarkably high resistance, reaching as high as 38.5% over a period of one year [38]. In other Asian regions, resistance has increased from 0 to 24.1% in 5 years [38,42], whereas in our isolates, it was only 0.5%. None of our isolates were detected with the presence of IS elements; however, 10% represented “silent *cfiA* carriers” as these isolates were phenotypically susceptible despite carrying the *cfiA* gene. 

In our study, the presence of the *cfiA* gene was only restricted to *B. fragilis* strains; based on that, we have categorized our *B. fragilis* strains into Division I (*cfiA*-negative) and Division II (*cfiA*-positive). This categorization holds a potential significance as MALDI-TOF MS can successfully differentiate these strains based on the well-defined mass spectra produced by *cfiA*-negative and -positive strains. The limitation is that MALDI-TOF MS can only identify the resistant genotype. As shown in a study, the IS acquisition and resultant carbapenem resistance were not differentiated via MALDI-TOF MS [43]. In our study, 42.5% of the *B. fragilis* isolates harbored silent *cfiA* genes and belonged to Division II; however, the unaccompanied presence appeared to be a handicap without insertion sequences. The results show that presently, imipenem resistance is not a challenge to current treatment options. However, these isolates may act as a reservoir of imipenem resistance since *B. fragilis* harbored a significant repertoire of the *cfiA* gene. With time, this may pose a threat, as susceptible strains may become resistant via the one-step acquisition of mobile IS elements from non-adjacent *cfiA* sites [12].

Another reason to study the distribution of *cfiA* genes is their association with a decrease in susceptibility to BL-BLIs such as piperacillin–tazobactam [44]. Though in our isolates, we determined that out of 27.5% (55/200) piperacillin–tazobactam-resistant isolates, 12.7% (7/55) carried the *cfiA* gene. We cannot comment on whether the presence of *cfiA* was responsible for seven piperacillin–tazobactam-resistant phenotypes, as IS elements required for the *cfiA* expression were absent, and such expression of the *cfiA* gene without IS elements is not reported in the literature. However, two studies have demonstrated that the strains showing heterogeneous resistance to carbapenems may exploit a weak self-promotor of the *cfiA* genes, which we think could be likely as our *cfiA*-positive strains showed reduced susceptibility towards imipenem as well [45,46]. However, piperacillin–tazobactam resistance could be attributed to the presence of other ß-lactam genes (*cfxA*) as six out of seven piperacillin–tazobactam-resistant phenotypes containing the *cfiA* gene contained the *cfxA* gene as well, but lacked *cepA*. 

In our study, piperacillin–tazobactam resistance showed a strong correlation with the presence of the *cepA* gene. *cepA* is ß-lactamase that can hydrolyze both penicillins and cephalosporins; however, it is frequently inhibited by ß-lactamase inhibitors [47]. Only 19 out of 200 clinical isolates showed the presence of *cepA*, which was limited to *B. fragilis* only. The *cepA* genes are known to provide both low- and high-level resistance to penicillins. The high-level resistance is attributable to the overexpression of the *cepA* gene due to the transcriptional activation via putative IS family IS*21* [48]. We did not study the *IS*21 elements in our isolates, but all isolates harboring the *cepA* gene exhibited piperacillin–tazobactam resistance; thus, we propose the presence of these elements in our isolates.

The *cfxA* gene was found in 29% (16/55) of piperacillin–tazobactam-resistant phenotypes and 18.6% (27/145) of susceptible phenotypes. These results are in line with those of previous studies which have reported the presence of *cfxA* in amoxicillin–clavulanate-resistant strains, thereby suggesting the differential expression of the *cfxA* gene [49]. Our results confirm a significant association between the *cfxA* gene and cefoxitin resistance; however, 11/53 isolates showed resistance but did not harbor the gene. This may suggest for other resistance mechanisms, e.g*.,* reduced penetration of the drug through the outer membranes, upregulation of drug efflux, presence of other ß-lactamases, accumulation of mutations in the outer membrane porin molecules, and penicillin-binding proteins [31,49]. 

The *cfxA* gene was seen widely distributed among non-*Bacteroides* species, explaining their role in conferring resistance to ß-lactams in other anaerobic isolates, as *cepA* and *cfiA* were somehow restricted to *B. fragilis* only. Furthermore, our results demonstrated a constrained co-existence of *cepA* and *cfiA* gene in *B. fragilis* isolates, as all the isolates positive with *cepA* gene lacked the *cfiA* gene and vice versa. Similar associations have been seen in other studies, confirming the acquisition of these two genes in unique and separate events. The origin of these genes has also been discriminated on the basis of different G+C mol% content and their localization, as *cepA* is chromosomal, whereas *cfiA* is located on both chromosomes and plasmids [45,50,51]. It is believed that the two divisions have remained continuously isolated to circumvent HGT, and gene transfer between these two divisions is unlikely [52]. Therefore, we confirmed whether any AMR gene was found to have an affinity to any of these divisions. No significant association of any other AMR gene was markedly noted with the members of Divisions I and II. 

In our study, all organisms showed 100% susceptibility to chloramphenicol. The absence of chloramphenicol resistance may reflect the occasional use of this drug in our clinical setup. For more than half of these isolates, the clustering of MICs was observed around the breakpoints (12% at 8 mg/L; 52% at 4 mg/L). Similar clustering was observed in another study as well [53], which may pose a threat in case of MIC creep over time. None of the tested isolates harbored chloramphenicol resistance genes, thereby confirming that they do not act as a reservoir of chloramphenicol resistance.

## 4. Materials and Methods

### 4.1. Population under Study

A total of 200 consecutive AGNB were collected from the clinical specimens received at the Clinical Bacteriology Laboratory of the department of Medical Microbiology. The sample size was decided based on the existing frequency of major AGNB to ensure a minimum of 30 isolates, the number recommended by CLSI (M39-A4), to obtain a reasonable statistical estimate of cumulative percent susceptibility rates or percentage calculations. The isolates were recovered from clinical specimens such as blood, pus, body fluids, necrotic tissues, abscess, peritoneal fluids, etc.

### 4.2. Sample Processing

On receiving a sample in the laboratory, the processing commenced with Gram staining followed by aerobic and anaerobic culture. In the anaerobic culture, the samples were inoculated in Robertson’s cooked meat (RCM) broth and freshly prepared brucella agar plates supplemented with 5% laked sheep blood, vitamin K (1 mg/L), hemin (5 mg/L), and a metronidazole (5 µg) disk placed at the center of the plate for the presumptive presence of anaerobes. An automated anaerobic gas generation system (Anoxomat^®^ Mart II, Mart Microbiology BV, Lichtenvooorde, The Netherlands) or Don Whitley Scientific’s Jar Gassing System^®^ were used to create anaerobiosis. In each run, the anaerobiosis was validated by biological indicators such as *B. fragilis* ATCC 25,285 and *Pseudomonas aeruginosa* ATCC 27,853 as positive and negative controls, respectively. Following incubation at 37 °C for 48 h, the anaerobic plates were examined for a zone of inhibition around the metronidazole disk. The presence of the zone, irrespective of the size, was suggestive of anaerobic growth. In case of no zone of inhibition, the anaerobic plates were correlated with their aerobic plates and turbidity in RCM to rule out metronidazole-resistant anaerobic growth. We identified the isolates using MALDI-TOF MS, Biotyper 2.0 database (Bruker Daltonik GmbH, Bremen, Germany) following the standard Bruker interpretative criteria. All isolates were correctly identified up to species level with a MALDI score value of ≥2.0.

### 4.3. Antimicrobial Susceptibility Testing

AST of these isolates was performed using the breakpoint agar dilution method as per the CLSI (M11-A8) protocol [10]. The antibiotics were chosen according to the CLSI recommended guidelines of antibiotic selection for AST. All antimicrobial agents recommended for the primary testing of anaerobes (metronidazole, clindamycin, imipenem, and piperacillin–tazobactam) were included, along with two supplemental agents recommended for the selective testing (cefoxitin and chloramphenicol).

A serial two-fold dilution of each antibiotic was added into brucella agar supplemented with 5% laked sheep blood, vitamin K (1 mg/L), and hemin (5 mg/L). An inoculum of 0.5 McFarland standard in enriched thioglycolate broth was prepared, and a 10 µL volume was spot inoculated on the surface of the freshly prepared plates to achieve a concentration of 10^5^ CFU/spot. A total of 25 isolates were tested on one plate. Following 48 h anaerobic incubation, the inoculated spots were examined visually for bacterial growth. The lowest concentration of antibiotics that prevented bacterial growth was the MIC. Similarly, if the growth persisted at the maximum concentration of the tested antibiotic, results were reported as greater than the highest concentration tested. The MIC was interpreted as per the CLSI breakpoints (Appendix A). In each run, *B. fragilis ATCC 25,285* was used as quality control of AST.

### 4.4. Genotypic Characterization and Detection of Antimicrobial Resistance Determinants

Using targeted PCR, isolates were tested for the presence of AMR genes conferring resistance to metronidazole (*nimE*), imipenem (*cfiA*), piperacillin–tazobactam (*cepA*), cefoxitin (*cfxA*), clindamycin (*ermF*), chloramphenicol (*cat*) and MGEs such as *cfiA^IS^* and IS*1186. B. fragilis* strains known to have AMR genes were used as a positive control. For the detection of *IS*1186, the positive control was provided by Dr. Joseph Soki, Institute of the Clinical Microbiology University of Szeged, Hungary. An overview of the PCR primers and the thermal cycling parameters is mentioned in Appendix A [3,54,55,56,57,58], and the reaction setup is summarized in Appendix A. 

### 4.5. DNA Extraction

The DNA was extracted using QIAamp DNA Kits (Qiagen) as per the manufacturer’s protocol for Gram-negative bacteria. The DNA concentrations and purity were determined using a Nanodrop^®^ spectrophotometer (Thermo Fisher Scientific, Waltham, Massachusetts, U.S.). Strains were stored at −80 °C in brain heart infusion broth (BHI) with 40% (*v/v*) glycerol for future reference [10]. 

### 4.6. Statistical Analysis

GraphPad Prism (v.9.0) and MedCalc (v.19.8) were used to study the statistical analyses, agreements, and correlations. An unpaired t-test was used to compare discrete variables. The categorical variables were present as percentages and frequencies. The Chi-square test and the Fisher exact test were used to assess the associations and significant differences between the presence of antimicrobial-resistant determinants among susceptible and not susceptible strains. Statistical significance was set at a *p*-value ≤ 0.05, and all values were two-tailed.

## 5. Conclusions

This study was planned to explore the role of human clinical AGNB as a reservoir of AMR genes, and it concludes that AGNB prevalent in our clinical setup may act as a reservoir of specific AMR genes. Thus, anti-anaerobic drugs must be used judiciously, and the pace of empirical management of anaerobic infections needs to be decelerated. 

## Figures and Tables

**Figure 1 antibiotics-12-00942-f001:**
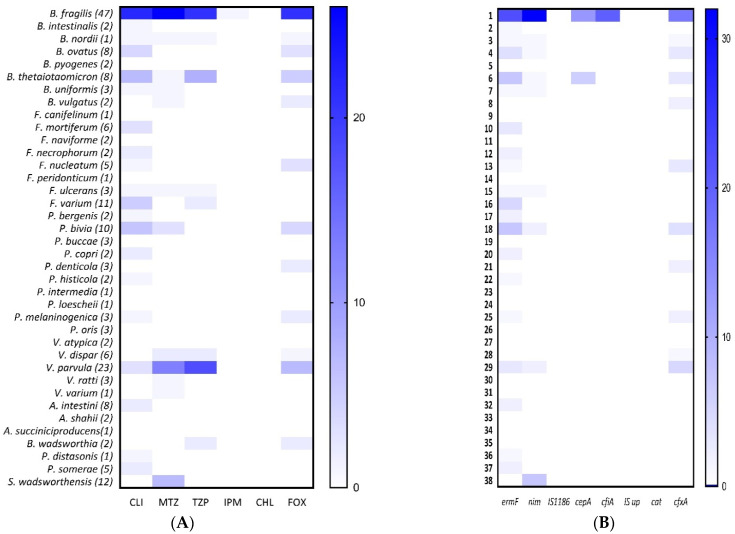
(**A**) Prevalence of antimicrobial resistance in clinical anaerobic Gram-negative bacteria. (**B**) Prevalence of antimicrobial resistance genes. The heat map depicts the association between resistant phenotypes and genotypes.

**Figure 2 antibiotics-12-00942-f002:**
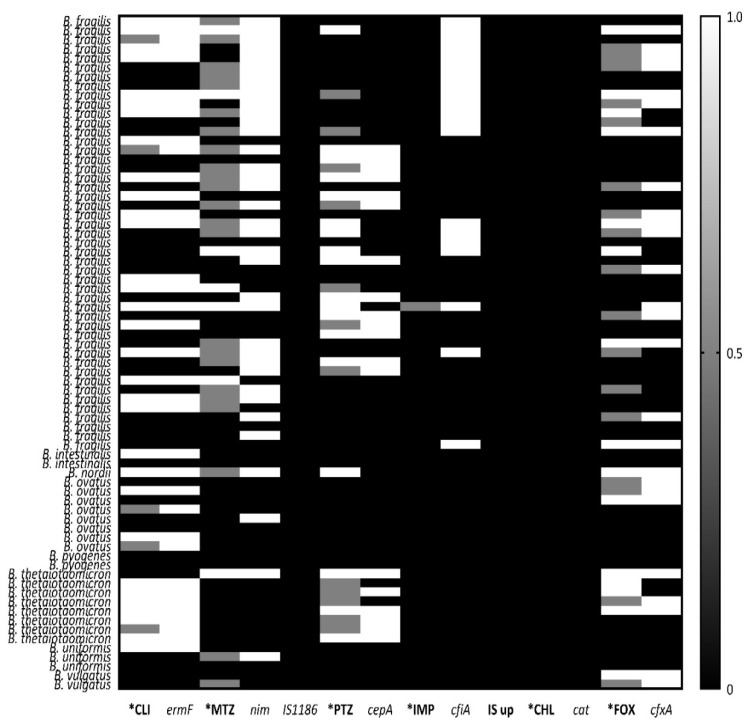
Prevalence of AMR, AMR genes and the association between resistant phenotypes and genotypes in *Bacteroides* spp. The white color and score 1 signify the presence of the gene and resistant phenotype; grey color and score 0.5 signify intermediate phenotype; black color and score 0 signify the absence of the gene and susceptible phenotype.

**Figure 3 antibiotics-12-00942-f003:**
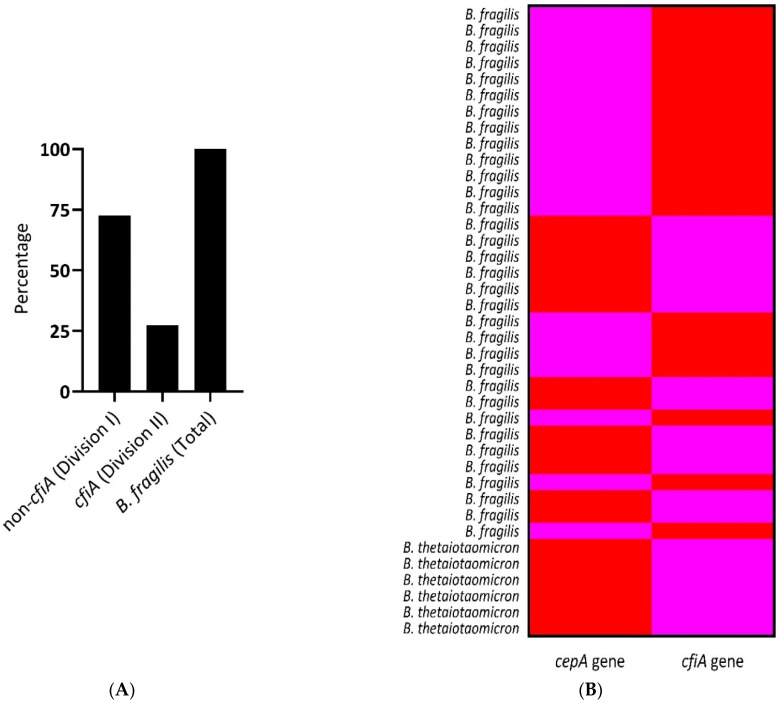
Distribution of *Bacteroides fragilis* group. (**A**) Prevalence of Division I (72.6%) and Division II (27.3%) based on the absence and presence of the *cfiA* gene, respectively. (**B**) The figure demonstrates the constrained co-existence of *cepA* and *cfiA* gene in *B. fragilis* species. (Red: presence of the gene; Pink: absence of the gene).

**Table 1 antibiotics-12-00942-t001:** Distribution of clinical anaerobic Gram-negative bacteria included in the study.

Genus (*n* = 11)	Species (*n* = 38)	(*n* = 200)	Percentage
Anaerobic Gram-Negative Bacilli (*n* = 157; 78.5%)			
*Bacteroides* spp. (*n* = 73; 36.5%)	*B. fragilis*	47	23.5%
	*B. intestinalis*	2	1.0%
	*B. nordii*	1	0.5%
	*B. ovatus*	8	4.0%
	*B. pyogenes*	2	1.0%
	*B. thetaiotaomicron*	8	4.0%
	*B. uniformis*	3	1.5%
	*B. vulgatus*	2	1.0%
*Fusobacterium* spp. (*n* = 31; 15.5%)	*F. canifelinum*	1	0.5%
	*F. mortiferum*	6	3.0%
	*F. naviforme*	2	1.0%
	*F. necrophorum*	2	1.0%
	*F. nucleatum*	5	2.5%
	*F. peridonticum*	1	0.5%
	*F. ulcerans*	3	1.5%
	*F. varium*	11	5.5%
*Prevotella* spp. (*n* = 30; 15%)	*P. bergensis*	2	1.0%
	*P. bivia*	10	5.0%
	*P. buccae*	3	1.5%
	*P. copri*	2	1.0%
	*P. denticola*	3	1.5%
	*P. histicola*	2	1.0%
	*P. intermedia*	1	0.5%
	*P. loescheii*	1	0.5%
	*P. melaninogenica*	3	1.5%
	*P. oris*	3	1.5%
*Alistipes* spp. (*n* = 2; 1%)	*A. shahii*	2	1.0%
*Anaerobiospirillum* spp. (*n* = 1; 0.5%)	*A. succiniciproducens*	1	0.5%
*Bilophilia* spp. (*n* = 2; 1%)	*B. wadsworthia*	2	1.0%
*Parabacteroides* spp. (*n* = 1; 0.5%)	*P. distasonis*	1	0.5%
*Porphyromonas* spp. (*n* = 5; 2.5%)	*P. somerae*	5	2.5%
*Sutterella* spp. (*n* = 12; 6%)	*S. wadsworthensis*	12	6.0%
Anaerobic Gram-Negative Cocci (*n* = 43; 21.5%)			
*Acidaminococcus* spp. (*n* = 8; 4%)	*A. intestini*	8	4.0%
*Veillonella* spp. (*n* = 35; 17.5%)	*V. atypica*	2	1.0%
	*V. dispar*	6	3.0%
	*V. parvula*	23	11.5%
	*V. ratti*	3	1.5%
	*V. varium*	1	0.5%

**Table 2 antibiotics-12-00942-t002:** The number of not susceptible (I+R) isolates with the percentage resistance against selected antimicrobials and the occurrence of corresponding resistance determinants in the tested species of genus *Bacteroides* (*n* = 73).

Species	No. of Resistant Strains (*n*)						Prevalence of Antimicrobial Resistance Genes (*n*)							
	CLI	MTZ	TZP	IPM	CHL	FOX	*ermF*	*nim*	*IS*1186	*cepA*	*cfiA*	*IS* up	*cat*	*cfxA*
Breakpoint mg/L	≥4	≥16	≥32/4	≥8	≥16	≥32								
*B. fragilis* (47)	22	26	21	1	0	21	22	32	0	13	20	0	0	17
*B. intestinalis* (2)	1	0	0	0	0	0	1	0	0	0	0	0	0	0
*B. nordii* (1)	1	1	1	0	0	1	1	1	0	0	0	0	0	1
*B. ovatus* (8)	4	0	0	0	0	3	4	1	0	0	0	0	0	3
*B. pyogenes* (2)	0	0	0	0	0	0	0	0	0	0	0	0	0	0
*B. thetaiotaomicron* (8)	7	1	8	0	0	5	7	1	0	6	0	0	0	3
*B. uniformis* (3)	1	1	0	0	0	0	1	1	0	0	0	0	0	0
*B. vulgatus* (2)	0	1	0	0	0	2	0	0	0	0	0	0	0	2

**Table 3 antibiotics-12-00942-t003:** The number of not susceptible (I+R) isolates with the percentage resistance against selected antimicrobials and the occurrence of corresponding resistance determinants in the tested species of genus *Fusobacterium* (*n* = 31).

Species	No. of Resistant Strains (*n*)						Prevalence of Antimicrobial Resistance Genes (*n*)							
	CLI	MTZ	TZP	IPM	CHL	FOX	*ermF*	*nim*	*IS*1186	*cepA*	*cfiA*	*IS* up	*cat*	*cfxA*
Breakpoint mg/L	≥4	≥16	≥32/4	≥8	≥16	≥32								
*F. canifelinum* (1)	0	0	0	0	0	0	0	0	0	0	0	0	0	0
*F. mortiferum* (6)	3	0	0	0	0	0	3	0	0	0	0	0	0	0
*F. naviforme* (2)	0	0	0	0	0	0	0	0	0	0	0	0	0	0
*F. necrophorum* (2)	2	0	0	0	0	0	2	0	0	0	0	0	0	0
*F. nucleatum* (5)	1	0	0	0	0	3	1	0	0	0	0	0	0	3
*F. peridonticum* (1)	0	0	0	0	0	0	0	0	0	0	0	0	0	0
*F. ulcerans* (3)	1	1	1	0	0	0	1	1	0	0	0	0	0	0
*F. varium* (11)	5	0	2	0	0	0	5	0	0	0	0	0	0	0

**Table 4 antibiotics-12-00942-t004:** The number of not susceptible (I+R) isolates with the percentage resistance against selected antimicrobials and the occurrence of corresponding resistance determinants in the tested species of genus *Prevotella* (*n* = 30).

Species	No. of Resistant Strains (*n*)						Prevalence of Antimicrobial Resistance Genes (*n*)							
	CLI	MTZ	TZP	IPM	CHL	FOX	*ermF*	*nim*	*IS*1186	*cepA*	*cfiA*	*IS* up	*cat*	*cfxA*
Breakpoint mg/L	≥4	≥16	≥32/4	≥8	≥16	≥32								
*P. bergenis* (2)	1	0	0	0	0	0	1	0	0	0	0	0	0	0
*P. bivia* (10)	6	3	0	0	0	4	6	2	0	0	0	0	0	4
*P. buccae* (3)	0	0	0	0	0	0	0	0	0	0	0	0	0	0
*P. copri* (2)	2	0	0	0	0	0	2	0	0	0	0	0	0	0
*P. denticola* (3)	0	0	0	0	0	2	0	0	0	0	0	0	0	2
*P. histicola* (2)	1	0	0	0	0	0	1	0	0	0	0	0	0	0
*P. intermedia* (1)	0	0	0	0	0	0	0	0	0	0	0	0	0	0
*P. loescheii* (1)	0	0	0	0	0	0	0	0	0	0	0	0	0	0
*P. melaninogenica* (3)	1	0	0	0	0	2	1	0	0	0	0	0	0	2
*P. oris* (3)	0	0	0	0	0	0	0	0	0	0	0	0	0	0

**Table 5 antibiotics-12-00942-t005:** The number of not susceptible (I+R) isolates with the percentage resistance against selected antimicrobials and the occurrence of corresponding resistance determinants in the tested species of genus *Veillonella* (*n* = 35).

Species	No. of Resistant Strains (*n*)						Prevalence of Antimicrobial Resistance Genes (*n*)							
	CLI	MTZ	TZP	IPM	CHL	FOX	*ermF*	*nim*	*IS*1186	*cepA*	*cfiA*	*IS* up	*cat*	*cfxA*
Breakpoint mg/L	≥4	≥16	≥32/4	≥8	≥16	≥32								
*V. atypica* (2)	0	0	0	0	0	0	0	0	0	0	0	0	0	0
*V. dispar* (6)	0	2	2	0	0	1	0	0	0	0	0	0	0	1
*V. parvula* (23)	3	13	18	0	0	7	3	2	0	0	0	0	0	5
*V. ratti* (3)	0	1	0	0	0	0	0	0	0	0	0	0	0	0
*V. varium* (1)	0	1	0	0	0	0	0	0	0	0	0	0	0	0

**Table 6 antibiotics-12-00942-t006:** The number of not susceptible (I+R) isolates with the percentage resistance against selected antimicrobials and the occurrence of corresponding resistance determinants in various other Gram-negative anaerobes (*n* = 31).

Species	No. of Resistant Strains (*n*)						Prevalence of Antimicrobial Resistance Genes (*n*)							
	CLI	MTZ	TZP	IPM	CHL	FOX	*ermF*	*nim*	*IS*1186	*cepA*	*cfiA*	*IS* up	*cat*	*cfxA*
Breakpoint mg/L	≥4	≥16	≥32/4	≥8	≥16	≥32								
*A. intestini* (8)	2	0	0	0	0	0	2	0	0	0	0	0	0	0
*A. shahii* (2)	0	0	0	0	0	0	0	0	0	0	0	0	0	0
*A. succiniciproducens* (1)	0	0	0	0	0	0	0	0	0	0	0	0	0	0
*B. wadsworthia* (2)	0	0	2	0	0	2	0	0	0	0	0	0	0	0
*P. distasonis* (1)	1	0	0	0	0	0	1	0	0	0	0	0	0	0
*P. somerae* (5)	2	0	0	0	0	0	2	0	0	0	0	0	0	0
*S. wadsworthensis* (12)	0	7	0	0	0	0	0	7	0	0	0	0	0	0

**Table 7 antibiotics-12-00942-t007:** The overall susceptibility results and the distribution of resistance determinants in clinical anaerobic Gram-negative bacteria included in the study.

Drug	No.	R (%)	*Bacteroides* spp. (*n* = 73)	*Fusobacterium* spp. (*n* = 31)	*Prevotella* spp. (*n* = 30)	*Veillonella* spp. (*n* = 35)	Others (*n* = 31)
CLI	67	(33.5%)	36 (49.3%)	12 (38.7%)	11 (36.7%)	3 (8.6%)	5 (16.1%)
MTZ	58	(29%)	30 (41.1%)	1 (3.2%)	3 (10%)	17(48.6%)	7 (22.6%)
TZP	55	(27.5%)	30 (41.1%)	3 (9.7)	0 (0%)	20 (57.1%)	2 (6.5%)
IPM	1	(0.5%)	1 (1.4%)	0 (0%)	0 (0%)	0 (0%)	0 (0%)
CHL	0	(0%)	0 (0%)	0 (0%)	0 (0%)	0 (0%)	0 (0%)
FOX	53	(26.5%)	32 (43.8%)	3 (9.7%)	8 (26.7%)	8 (22.9%)	2 (6.5%)
Genes	No.	R (%)	*Bacteroides* spp. (*n* = 73)	*Fusobacterium* spp. (*n* = 31)	*Prevotella* spp. (*n* = 30)	*Veillonella* spp. (*n* = 35)	Others (*n* = 31)
*ermF*	67	(33.5%)	36 (49.3%)	12 (38.7%)	11 (36.6%)	3 (8.6%)	5 (16.1%)
*nim*	48	(24%)	36 (49.3%)	1 (3.2%)	2 (6.7%)	2 (5.7%)	7 (22.6%)
*IS*1186	0	(0%)	0 (0%)	0 (0%)	0 (0%)	0 (0%)	0 (0%)
*cepA*	19	(9.5%)	19 (26.0%)	0 (0%)	0 (0%)	0 (0%)	0 (0%)
*cfiA*	20	(10%)	20 (27.4%)	0 (0%)	0 (0%)	0 (0%)	0 (0%)
*IS* up	0	(0%)	0 (0%)	0 (0%)	0 (0%)	0 (0%)	0 (0%)
*cat*	0	(0%)	0 (0%)	0 (0%)	0 (0%)	0 (0%)	0 (0%)
*cfxA*	43	(21.5%)	26 (35.6%)	3 (9.7%)	8 (26.7%)	6 (17.1%)	0 (0%)

**Table 8 antibiotics-12-00942-t008:** Agreement between phenotypic and genotypic resistance in clinical Gram-negative anaerobic isolates.

Antimicrobial Agent	Association of Antimicrobial Resistant Phenotypes and Genotypes										
	Genes	PR	GR	PR+/GR+	PR−/GR−	PR+/GR−	PR−/GR+	Agreement	*IS*1186	*cfiA^IS^*	Agreement
								*p*-Value			*p*-Value
Clindamycin	*ermF*	67	67	67	133	0	0	<0.05	0	0	<0.05
Metronidazole	*nimE*	58	48	35	129	23	13	<0.05	0	0	<0.05
Piperacillin–tazobactam	*cepA*	55	19	19	145	36	0	<0.05	0	0	<0.05
Imipenem	*cfiA*	1	20	1	180	0	19	>0.9999	0	0	<0.05
Chloramphenicol	*cat*	0	0	0	200	0	0	<0.05	0	0	<0.05
Cefoxitin	*cfxA*	53	43	42	146	11	1	<0.05	0	0	<0.05

PR, number of isolates expressing phenotypic resistance to the indicated antimicrobial agent; GR, number of isolates carrying the indicated antimicrobial resistance gene; PR+/GR+, phenotypically resistant isolates carrying antimicrobial resistance genes; PR−/GR−, phenotypically susceptible isolates carrying no antimicrobial resistance genes; PR+/GR−, phenotypically resistant isolates carrying no antimicrobial resistance genes; PR−/GR+, phenotypically susceptible isolates carrying antimicrobial resistance genes; IS*1186*, isolates carrying insertion sequence *1186; cfiA^IS^*, *cfiA* gene (350 bp), and the intact segment containing *cfiA* gene and upstream IS elements (1.6–1.7 kb).

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
