# Peer review of "Anaerobic Gram-Negative Bacteria: Role as a Reservoir of Antibiotic Resistance"

_antibiotics, 2023, doi:10.3390/antibiotics12050942_

Round 1

Reviewer 1 Report

I don't think you have to do anything in particular to publish it, good presentation, good explanation and good presentation of results. I hope they will do a similar study with a greater number of isolates.

Author Response

Thank you for taking the time to review our manuscript and for your positive feedback. We appreciate your suggestion about conducting a similar study with a greater number of isolates, and we agree that this would be an interesting avenue for future research. However, we hope that our current study will provide a valuable contribution to the understanding of antimicrobial resistance in Gram-negative anaerobic bacteria. Thank you again for your comments and suggestions.

Reviewer 2 Report

The research that is described in this manuscript is well done; nevertheless, the reviewer has several non-critical remarks and one important question. Firstly, in the "Introduction" section, there are sorely lacking references to the literature. Then, on lines 426-428, there is the text “Please turn to the CRediT taxonomy for the term explanation. Authorship must be limited to those who have contributed substantially to the work reported.” It seems that it should be removed, probably from the template for preparing the manuscript.

In this manuscript, the authors show really important results. But there is a question about the design of the study.

In the "introduction" section, the authors write that the intestinal microflora (almost regardless of the relationship of bacteria to oxygen) is a potential reservoir for antibiotic resistance genes, apparently, it is so. However, in the study itself, the authors used anaerobic Gram-negative bacteria that were isolated from clinical samples (lines 356-357 indicate blood, pus, necrotic tissue, abscess, peritoneal fluid, etc.). Are these clinical isolates of intestinal origin or are they hospital strains? Is there any evidence for this? Maybe it would be necessary to compare the distribution of genotypic and phenotypic markers of antibiotic resistance in clinical isolates with the same microbes isolated from the intestine? Or maybe it's better to remake the introduction somehow? In essence, the manuscript deals with the prevalence of antibiotic resistance among clinical isolates, and how the phenotypic manifestation of antibiotic resistance is due to the genotype.

Author Response

Response to Reviewer 2 Comments

The research that is described in this manuscript is well done; nevertheless, the reviewer has several non-critical remarks and one important question.

Point 1: Firstly, in the "Introduction" section, there are sorely lacking references to the literature.

Response 1: Thank you for bringing this to our attention, and we appreciate your valuable input in improving the manuscript. We apologize for the lack of references in the "Introduction" section. We have reviewed the existing literature on the topic and ensured that appropriate references are added to provide a comprehensive background and support for our study.

Point 2: Then, on lines 426-428, there is the text “Please turn to the CRediT taxonomy for the term explanation. Authorship must be limited to those who have contributed substantially to the work reported.” It seems that it should be removed, probably from the template for preparing the manuscript.

Response 2: We apologize for the oversight in leaving the lines "Please turn to the CRediT taxonomy for the term explanation. Authorship must be limited to those who have contributed substantially to the work reported." in the manuscript. We acknowledge that these lines were not intended to be included and have now been removed.

Point 3: In this manuscript, the authors show really important results. But there is a question about the design of the study. In the "introduction" section, the authors write that the intestinal microflora (almost regardless of the relationship of bacteria to oxygen) is a potential reservoir for antibiotic resistance genes, apparently, it is so. However, in the study itself, the authors used anaerobic Gram-negative bacteria that were isolated from clinical samples (lines 356-357 indicate blood, pus, necrotic tissue, abscess, peritoneal fluid, etc.). Are these clinical isolates of intestinal origin or are they hospital strains? Is there any evidence for this? Maybe it would be necessary to compare the distribution of genotypic and phenotypic markers of antibiotic resistance in clinical isolates with the same microbes isolated from the intestine? Or maybe it's better to remake the introduction somehow? In essence, the manuscript deals with the prevalence of antibiotic resistance among clinical isolates, and how the phenotypic manifestation of antibiotic resistance is due to the genotype.

Response 3: We appreciate your valuable feedback and thoughtful comments regarding the design of our study and the information presented in the introduction section. In our study, we focused on anaerobic Gram-negative bacteria (AGNB) isolated from clinical samples such as blood, pus, necrotic tissue, abscess, peritoneal fluid, etc. These clinical isolates were obtained from patients with various infections, and while they are not specifically of intestinal origin, they can contribute valuable insights into the prevalence of antibiotic resistance among clinical isolates. We agree that further studies comparing the resistance profiles of clinical isolates with those from the intestine would be of interest and could provide additional insights into the relationship between the phenotype and genotype of antibiotic resistance. However, such comparisons were not the primary focus of our study. We completely agree with you that in essence, the manuscript deals with the “prevalence of antibiotic resistance among clinical isolates, and how the phenotypic manifestation of antibiotic resistance is due to the genotype”. Therefore we have revised the introduction to shift the focus from the gut microbiota to clinically significant anaerobic Gram-negative bacteria (AGNB) isolated from clinical samples. This adjustment aligns the manuscript with our study's objective of exploring the prevalence of antimicrobial resistance (both genotypic and phenotypic) among AGNB in clinical settings, their role as reservoirs of antibiotic resistance, and the need for investigating their AMR patterns. We believe the revised introduction addresses the raised concerns. Thank you for your valuable input, this has contributed to the improvement of our manuscript.

Also here I am attaching the revised introduction:

“Introduction: The emerging antimicrobial resistance (AMR) is a critical problem faced by the medical and scientific community. While extensive research has addressed AMR in aerobic and facultative anaerobic pathogens, the study on strict anaerobes has received comparatively less attention. Anaerobic Gram-negative bacteria (AGNB), comprise a significant proportion of the human microbiota and often act as secondary pathogens.[1] AGNB are the most common anaerobes associated with infections and include some of the most antimicrobial resistant species.[2] These bacteria, particularly the Bacteroides fragilis group, have exhibited notable resistance rates[3], with some strains demonstrating multiple-drug resistance (MDR).[4] MDR is now being recognized among other clinical AGNB as well, which were earlier believed to be susceptible.[5] Most of the available literature on AMR belongs to the genus Bacteroides, and the remaining AGNB continued to take a back seat. These bacteria can serve as channels for horizontal gene transfer (HGT) and the ignored conduct may lead to the possibility of selection and transfer of resistance determinants.[6]

Understanding the extent of AMR in AGNB is essential for several reasons. AGNB are clinically significant, often associated with polymicrobial infections, and can cause severe infections such as abscesses, bacteremia, and intra-abdominal infections.[2] The appropriate management of these infections requires accurate antimicrobial susceptibility testing (AST) and an understanding of local resistance patterns. However, current knowledge of AMR patterns in AGNB is limited, and existing data often deviate from standard guidelines.[7,8] The changing antibiograms and the emergence of resistance determinants in AGNB mandate periodic phenotypic and genotypic AST of these bacteria. Moreover, AGNB exhibit intrinsic resistance mechanisms that differ from those observed in aerobic bacteria. They possess distinct sets of resistance genes, including those encoding resistance to critical antibiotics[9], making effective treatment challenging. In AGNB, there has been a focus on studying AMR, especially genotypic resistance in limited species and against a restricted spectrum of antibiotics, many of which hold less therapeutic significance today.

 In this study, we aim to address these knowledge gaps and explore the prevalence of AMR among clinically significant AGNB. We assessed the phenotypic and genotypic resistance among clinically significant AGNB to at least one drug from each class of antimicrobial agents recommended by CLSI for the primary testing of anaerobes viz. metronidazole, clindamycin, piperacillin-tazobactam, and imipenem; and to two supplemental drugs for the selective testing viz. chloramphenicol and cefoxitin. Due to the comparable clinical efficacy and interpretive results, only one antimicrobial agent from each antimicrobial class was included.[4] We intentionally excluded all ß-lactams since most of the B. fragilis group members are reported as uniformly resistant to them.[5] In this study, the resistance to each antimicrobial agent was considered (a) Phenotypic resistance, an isolate with a “resistant” phenotype; when the MIC was greater than the breakpoint. The data is interpreted by including intermediate and resistant in the resistant category. (b) Genotypic resistance, an isolate with a “resistant” genotype; when the isolate harbored the gene encoding for AMR to the given antimicrobial agent. We also studied the prevalence of the AMR determinants for the afore-mentioned antimicrobials viz. nimE (metronidazole), cfiA (imipenem), cepA (piperacillin-tazobactam), cfxA (cefoxitin), ermF (clindamycin), cat (chloramphenicol) and MGEs like cfiAIS, insertion sequence present upstream of cfiA gene known to upregulate the expression of the cfiA gene and resultant imipenem resistance and, IS1186 associated with the cfiA expression and in many cases known to induce nim gene-mediated metronidazole resistance.

Overall, the study findings may have important implications, with the potential to enhance clinical decision-making, inform antibiotic stewardship efforts, and shape infection control strategies tailored to AGNB-associated infections. It underscores the importance of addressing AMR challenges and improving patient outcomes.”

Reviewer 3 Report

In this manuscript, the authors have looked into the phenotypes and genotypes of the clinically relevant anaerobic gram negative bacteria that might act as a reservoir of genes leading to antimicrobial resistance. They have reported increased resistance to clindamycin in Bacteroides spp. in association with ermF gene which can potentially be a reservoir for imparting resistance. They also found that all tested organisms showed complete susceptibility to chloramphenicol as well as high susceptibility to imipenem so far. The biggest conclusion that was made from this study was the absence of cepA gene when cfiA gene was present and vice versa in B. fragilis

Line 27,28: Needs to be rephrased. Complete association is unclear in the abstract unless the reader goes through the complete text.

Line 30 : Rephrase "constraint constrained"

Line 51,52,53: What are you referring to "these pathogens"? are you implying that the anaerobic gut flora can become pathogenic?

Line 81: Replace "taking" with "including"

Line 121: number of "non"-susceptible

Line 160-170: Reformatting required

Line 277, 278: When you are mentioning about overall carbepenem resistance as per western studies, what percentage of that reflects carbepenem resistance in gram negative anaerobes?

Line 297: "mobile" instead of "mobilizable"

Line 306: Please include appropriate reference for " weak self promotor" evidence.

Line 317, 318, 320: Needs reformatting

Overall, the study conducted is thorough and content wise, it is a well written manuscript .

Author Response

Response to Reviewer 3 Comments

In this manuscript, the authors have looked into the phenotypes and genotypes of the clinically relevant anaerobic gram negative bacteria that might act as a reservoir of genes leading to antimicrobial resistance. They have reported increased resistance to clindamycin in Bacteroides spp. in association with ermF gene which can potentially be a reservoir for imparting resistance. They also found that all tested organisms showed complete susceptibility to chloramphenicol as well as high susceptibility to imipenem so far. The biggest conclusion that was made from this study was the absence of cepA gene when cfiA gene was present and vice versa in B. fragilis

Point 1: Line 27,28: Needs to be rephrased. Complete association is unclear in the abstract unless the reader goes through the complete text. Line 27,28: The association between resistant phenotypes and genotypes was complete in clindamycin and chloramphenicol, whereas it was low among imipenem and piperacillin-tazobactam.

Response 1: Thank you for your suggestion to improve the clarity of these sentences. We have made the following revisions, “The association between resistant phenotypes and genotypes was complete in clindamycin, as all clindamycin resistant isolates showed the presence of ermF gene, and none of the susceptible strains harbored this gene; similarly all isolates were chloramphenicol susceptible and also lacked cat gene. Whereas the association was low among imipenem and piperacillin-tazobactam.

Point 2: Line 30 : Rephrase "constraint constrained"

Response 2: I apologize for the typo, it has been modified to “constrained”.

Point 3: Line 51,52,53: What are you referring to "these pathogens"? are you implying that the anaerobic gut flora can become pathogenic? Line 51,52,53: Studying the extent of AMR in anaerobic gut flora and its role as a reservoir of AMR genes can give important information leading to strategies for preventing and managing infections by these pathogens.

Response 3: Thank you for your insightful comment. Our intention was to highlight the potential for anaerobic gut flora, including Gram-negative anaerobic bacteria, predominantly present in the gut to become opportunistic pathogens and cause infections, especially in immunocompromised individuals. However, upon reflection and considering your remarks, we further refined our introduction, we have shifted the focus to clinically significant anaerobic Gram-negative bacteria (AGNB) isolated from clinical samples, thereby minimizing the unnecessary attention given to the gut microbiota. This adjustment clarifies our objective of studying the prevalence of antimicrobial resistance (AMR) among AGNB in clinical settings. We appreciate your feedback, which has contributed to the improvement of the manuscript's clarity and focus.

Point 4: Line 81: Replace "taking" with "including" Line 81: The data is interpreted by taking intermediate and resistant in the resistant category.

Response 4: Thank you, the sentence has been modified as “the data is interpreted by including intermediate and resistant in the resistant category.”

Point 5: Line 121: number of "non"-susceptible. Line 121: The number of not-susceptible isolates with the percentage resistance against selected antimicrobials and the occurrence of corresponding AMR genes in tested species of genus Bacteroides, Fusobacterium, Prevotella, Veillonella, and other AGNB are mentioned in table 2-6, respectively.

Response 5: We appreciate your attention to detail. However, we believe that our usage of 'not-susceptible' is appropriate in this context. We would like to clarify that. The term 'not-susceptible' aligns with the definitions provided by authoritative sources, including the Clinical and Laboratory Standards Institute (CLSI) and the European Committee on Antimicrobial Susceptibility Testing (EUCAST). We have consulted the publication by Kahlmeter et al. (2019) [Reference: PMID: 31315957; PMCID: PMC6711922], which discusses the differences in reporting antimicrobial susceptibility results. In accordance with these references, 'not-susceptible' is a category used for isolates for which only a susceptible breakpoint is designated because of the absence or rare occurrence of resistant strains. Isolates for which the antimicrobial agent MICs are above the susceptible breakpoint or whose zone diameters are below those associated with the susceptible breakpoint should be reported as nonsusceptible. An isolate that is interpreted as non-susceptible does not necessarily mean that the isolate has a resistance mechanism. It is possible that isolates for which MICs are above the susceptible breakpoint and that lack resistance mechanisms may be encountered within the wild-type distribution subsequent to the time that the susceptible-only breakpoint was set. The term “non-susceptible” should not be used when describing an organism/drug category with intermediate and resistant interpretive categories. Isolates that are in the categories of “intermediate” or “resistant” may be called “not-susceptible” rather than “non-susceptible.”

Point 6: Line 160-170: Reformatting required

Response 6: Thank you for bringing to our attention the issue regarding the use of italics in these sentences. We have made the changes.

Point 7: Line 277, 278: When you are mentioning about overall carbapenem resistance as per western studies, what percentage of that reflects carbapenem resistance in gram negative anaerobes? Line 277, 278: Western data from various studies have reported an overall carbapenem resistance varying from 1 to 9.6% and cfiA positivity of 5 to 27%.[41]

Response 7: We apologize for any confusion caused by the lack of specific mention of Gram-negative anaerobes in that particular sentence. However, we want to emphasize that the percentage and the data we cited specifically pertains to B. fragilis. To address your concern we have mentioned B. fragilis in the sentence. The new sentence reads as “Western data from various studies have reported an overall carbapenem resistance varying from 1 to 9.6% and cfiA positivity of 5 to 27% in B. fragilis.”

Point 8: Line 297: "mobile" instead of "mobilizable"

Response 8: Thank you, it has been replaced.

Point 9: Line 306: Please include appropriate reference for "weak self promotor" evidence.

Response 9: Upon reviewing our manuscript, we realized that the references supporting the statement were indeed included. However, there was an error in their placement, it somehow got placed in between the next sentence, which caused confusion. We sincerely apologize for this mistake. We have rectified the error and ensured that the references supporting the statement about "weak self-promoter" evidence are now appropriately placed within the manuscript as reference no. 45 and 46. We appreciate your diligence in pointing out this oversight, as it has allowed us to correct it promptly. I have also mentioned the following references below:

  1. Rasmussen, B.A.; Gluzman, Y.; Tally, F.P. Cloning and Sequencing of the Class B Beta-Lactamase Gene (CcrA) from Bacteroides Fragilis TAL3636. Antimicrob. Agents Chemother. 1990, 34, 1590, doi:10.1128/AAC.34.8.1590.
  2. Podglajen I; Breuil J; Casin I; Collatz E. Genotypic Identification of Two Groups within the Species Bacteroides Fragilis by Ribotyping and by Analysis of PCR-Generated Fragment Patterns and Insertion Sequence Content. J. Bacteriol. 1995, 177, 5270–5275, doi:10.1128/JB.177.18.5270-5275.1995.

Point 10: Line 317, 318, 320: Needs reformatting

Response 10: Thank you for pointing out the sentences written in bold. We have formatted it.

Point 11: Overall, the study conducted is thorough and content wise, it is a well written manuscript .

Response 11: Thank you for your positive feedback on our manuscript. We sincerely thank you for taking the time to review our manuscript and provide such encouraging feedback. Your comments are invaluable in helping us improve and strengthen our work.